# Functional Genomics Identified Novel Genes Involved in Growth at Low Temperatures in *Listeria monocytogenes*

Yansha Wu,[a] Xinxin Pang,[a] Xiayu Liu,[a] Yajing Wu,[a] ⓘ Xinglin Zhang[a,b]

[a]Department of Food Science and Nutrition, Zhejiang University, Hangzhou, China
[b]College of Agriculture and Forestry, Linyi University, Linyi, China

**ABSTRACT** *Listeria monocytogenes* (Lm) is a foodborne pathogen that can cause severe human illness. Standard control measures for restricting bacterial growth, such as refrigeration, are often inadequate as Lm grows well at low temperatures. To identify genes involved in growth at low temperatures, a powerful functional genomics method Tn-seq was performed in this study. This genome-wide screening comprehensively identified the known and novel genetic determinants involved in low-temperature growth. A novel gene *lmo1366*, encoding rRNA methyltransferase, was identified to play an essential role in Lm growth at 16°C. In contrast, the inactivation of *lmo2301*, a gene encoding the terminase of phage A118, significantly enhanced the growth of Lm at 16°C. The deletion of *lmo1366* or *lmo2301* resulted in cell morphology alterations and impaired the growth rate in milk and other conditions at low temperatures. Transcriptomic analysis revealed that the Δ*lmo1366* and Δ*lmo2301* mutants exhibited altered transcriptional patterns compared to the wild-type strain at 16°C with significant differences in genes involved in ribosome structural stability and function, and membrane biogenesis, respectively. This work uncovered novel genetic determinants involved in Lm growth at 16°C, which could lead to a better understanding of how bacteria survive and multiply at low temperatures. Furthermore, these findings could potentially contribute to developing novel antibacterial strategies under low-temperature conditions.

**IMPORTANCE** *Listeria monocytogenes* is a Gram-positive pathogen that contributes to foodborne outbreaks due to its ability to survive at low temperatures. However, the genetic determinants of Lm involved in growth at low temperatures have not been fully defined. Here, the genetic determinants involved in the low-temperature growth of Lm were comprehensively identified on a genome-wide scale by Tn-seq. The gene *lmo1366*, encoding rRNA methyltransferase, was identified essential for growth under low-temperature conditions. On the other hand, the gene *lmo2301*, encoding terminase of phage A118, plays a negative role in bacterial growth at low temperatures. The transcriptomic analysis revealed the potential mechanisms. These findings lead to a better understanding of how bacteria survive and multiply at low temperatures and could provide unique targets for novel antibacterial strategies under low-temperature conditions.

**KEYWORDS** *Listeria monocytogenes*, Tn-seq, lmo1366, lmo2301, low temperature, rRNA methyltransferase, ribosome

*L*isteria monocytogenes (Lm) is a foodborne pathogen that can cause severe human illness in susceptible patients, notably immunosuppressed individuals, pregnant women, and the elderly (1). Lm can cross human intestinal, blood-brain and fetal-placental barriers, leading to gastroenteritis, bacteremia, neurolisteriosis, miscarriage, or death (1–3). One retrospective study of patients in China reported 759 cases with 18% fatality rate, including a neonatal fatality rate as high as 73% (4).

Most human listeriosis cases appear to be caused by consuming foods contaminated with Lm (5, 6). While initial Lm levels in contaminated foods are usually low, the

Address correspondence to Xinglin Zhang, xinglinzhang@zju.edu.cn.

The authors declare no conflict of interest.

ability of Lm to survive and multiply at low temperatures allows it to reach levels high enough to cause human disease. Previous research has shown that Lm can grow in ready-to-eat vegetable salads stored at low temperatures (7). Although 4℃ is the most common temperature for refrigeration, a national survey of the United States showed that the home refrigeration temperatures for ready-to-eat foods could range from below 0℃ to 15.6℃ (8). There are also research reports on the survival or growth of Lm in low moisture foods (9), commercial pasteurized whole milk products (10), ready-to-eat meats (11), and ice cream (12) at 16℃. Therefore, there is a great need to under-stand the mechanisms of Lm growth at low temperatures.

Several methods, such as transcriptomics and proteomics, have been applied to explore the mechanisms contributing to the survival of Lm at cold temperatures, and great achievements have been made (13, 14). A previous review summarized the mechanisms contributing to Lm cold temperature survival and adaptation. Cold temperatures can reduce membrane fluidity, increase superhelical coiling of DNA, and secondary structures in RNA, affect translation, reduce enzyme activities, and cause inefficient or slow protein folding; ribosomes have to adapt to function correctly at low temperatures (15).

To further explore the genes of Lm involved in growth at low temperatures, transposon insertion sequencing (Tn-seq) was used. Tn-seq is a powerful method for determining con-ditionally essential regions in bacterial genomes (16, 17). Transposon sequencing requires constructing a high-density library with transposon insertions in most nonessential genes and screening under specific conditions. The relative frequency of each mutant in the pop-ulation of different experimental groups was determined by sequencing. From these data, it was possible to quantify the fitness contribution of each gene under each condition, identify essential regions, and infer the association of genes with specific phenotypes sys-temically (17). A previous study identified the nucleotide excision repair gene *uvrA* as the genetic determinant of desiccation tolerance of the *Streptococcus pneumoniae* by Tn-seq (18). Moreover, five novel *Streptococcus agalactiae* genes that promote survival in the pres-ence of human amniotic fluid were identified by Tn-seq (19).

Unlike RNA-Seq or other genomic studies technologies, Tn-seq screening is based on a library containing a large population of different mutants. In order to prevent false-positive results in subsequent sequencing and statistical analysis or overwhelm-ing the real key genes by the noncritical genes, relatively mild stress conditions were often used for Tn-seq screening (20, 21).

In this study, we chose 16℃ as the experimental temperature condition for Tn-seq screening, which is a trade-off between the realistic condition and the technical limitation. The high-density transposon mutant library of Lm was constructed previously in our lab, which contained 63666 unique insertion mutants (unpublished data). Following the Tn-seq screening, the Cre-*loxP* recombinase system was used to construct targeted gene deletion mutant strains, and mutant growth profiles at different temperatures, including 16℃ and 4℃ were characterized. Scanning electron microscopy (SEM) analysis was performed to observe morphological changes. Furthermore, transcriptomic analysis of wild-type and mu-tant strains at 16℃ was performed, which provided theoretical clues for mechanistic study.

## RESULTS AND DISCUSSION

**Screening for genes involved in low-temperature growth.** A high-density trans-poson mutant library of *Listeria monocytogenes* EGD-e was constructed in our previous study. The mutant library had more than 60,000 unique mutants, with an average of 46 bp per insertion. To identify genes involved in low-temperature growth, we per-formed Tn-seq on cultures of the Lm transposon mutant libraries upon growth at 16℃ and 37℃. On average, 10 million aligned reads were produced per sample. DNA plotter and Artemis were used to visualize the transposon insertion reads across the entire ge-nome. The results showed the distribution of TA sites transposon insertions and candi-date-essential genes, and no apparent Tn insertion bias was observed (Fig. 1A).

According to the mutant amount changes, the genes were classified into four groups (Fig. 1B). Housekeeping gene deletion mutants were incapable of surviving at 37℃ and

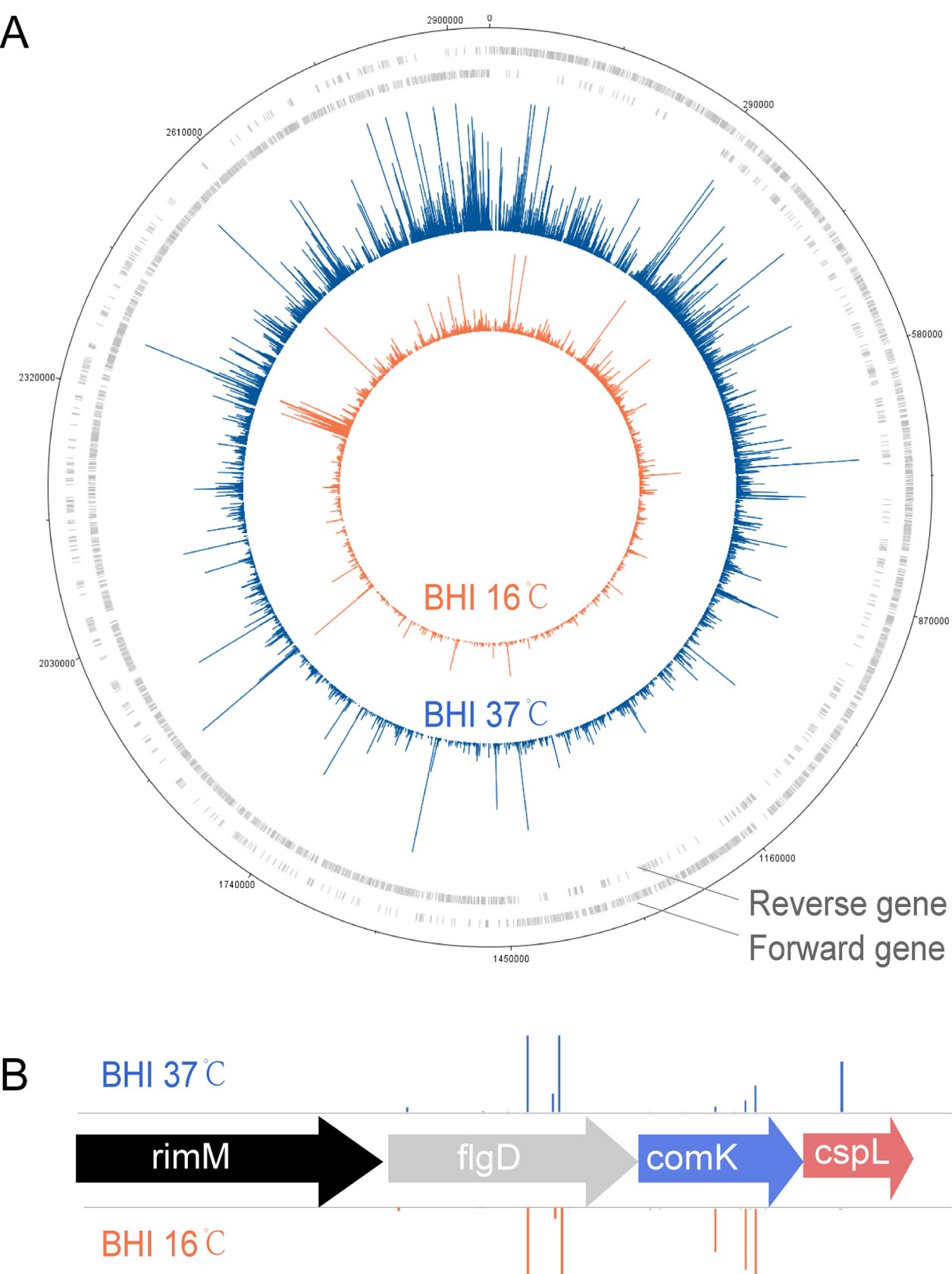

**FIG 1** Screening for genes involved in low temperature. (A) The tracks of the DNAPlotter map from the outside represent (1) Forward gene, (2) Reverse gene, (3) Insertion position of Tn and number of mutant survivals in BHI at 37°C; (4) Insertion position of Tn and number of mutant

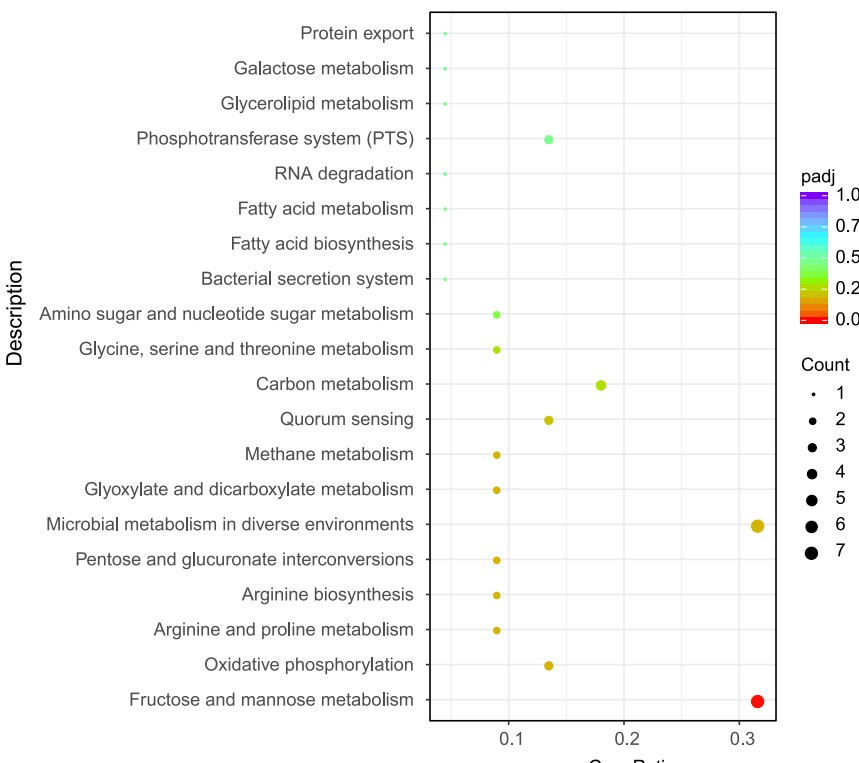

**FIG 2** Scatterplot of KEGG enrichment analysis. KEGG analysis results of enriched essential genes involved in growth in BHI at 16℃. The *y* axis represented enriched biological pathways; the *x* axis represented the ratio of differential genes to the total genes.

16℃ (black). For example, *rimM* (encoding 16S rRNA-processing protein) is essential at 37℃ and 16℃ (22). Nonessential-gene deletion mutants, such as insertions in *flgD* (encoding flagellar hook assembly protein), had no significant differences in growth at 37℃ and 16℃ (gray) (23). Similar to insertions in *comK*, negative-regulate-gene deletion mutants could survive better at 16℃ versus 37℃ (blue) (24). Conditional-essential-gene deletion mutants exhibited poor growth at 16℃ versus 37℃ (red). As proven in a previous study, the insertion in *cspL* resulted in a growth defect at low temperatures (25).

According to the Tn-seq results,61 gene deletion mutants survived better, and 79 gene deletion mutants survived worse at 16℃ significantly compared with 37℃ (|FoldChange|≥2, BH ≤ 0.01) (Table S1). KEGG analysis showed that the insertion of genes associated with fructose and mannose metabolism resulted in significantly worse survival of mutants (Fig. 2). The nutritional limitation affected the growth of Lm at low temperatures (26). To screen for candidate-essential genes for further study, we visualized the genes involved in low temperature with a bubble plot (Fig. 3).

Although some low-temperature-associated genes have been reported previously, many prospective genes have not been reported (Fig. 3). The most apparent genes *cspL* (encoding cold shock protein), *bipA* (encoding GTP-binding elongation factor homolog), *rhlA*, and *rhlD* (encoding DEAD/DEAH box helicase) had been knocked-out previously. The deletion mutants displayed a significant slow-growth phenotype at low temperatures (25). Cold shock protein enhanced the translation by blocking the development of secondary structures in mRNA (14). Furthermore, proteomics revealed

**FIG 1** Legend (Continued)
survival in BHI at 16℃. (B) Schematic representation of housekeeping gene (black, mutants cannot survive in both conditions), nonessential gene (gray, mutants have no effects on growth under the given conditions); negative-regulate gene (blue, mutants can survive better versus control); essential gene (red, mutants cannot grow well under the given conditions versus control); The height of column represents the survival number of the insertion mutants in BHI.

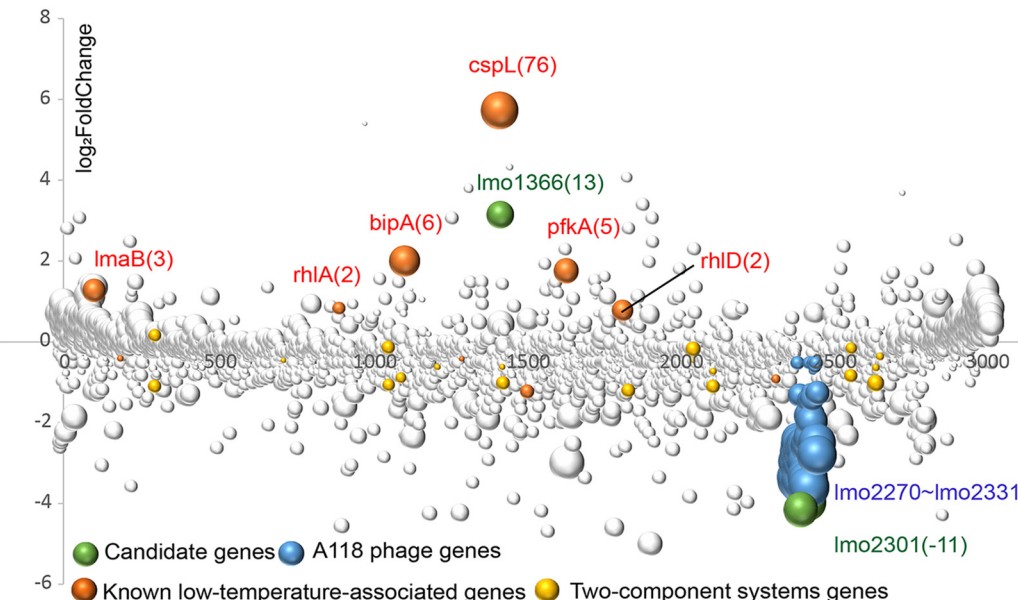

**FIG 3** Tn-seq results of genes involved in BHI at 16°C. The x axis represents the location of the gene in the chromosome, and the y axis represents the Log₂FoldChange value (37°C versus 16°C). Each bubble represents a gene, and the size of the bubble represents the BH values. The number besides gene represented the values of fold change. The positive values indicate that fewer mutants can survive at 16°C, while negative values indicate mutants have higher fitness at 16°C.

that CspL was overexpressed 65-fold at a low temperature (13). The GTP-binding elongation factor homolog could help translation elongation at low temperatures, and *bipA* was upregulated 8-fold at 4°C (27). DEAD/DEAH box RNA helicases helped relieve secondary structures formed by RNA at low temperatures, contributing to the ability of *L. monocytogenes* to grow at low temperatures (14). Furthermore, whole-genome microarray analysis identified higher transcript levels of genes encoding DEAD/DEAH box RNA helicases at 4°C compared to those grown at 37°C (27). Also, 6-phosphofructokinase encoded by *pfkA* catalyzes the formation of fructose 1 to 6-bisphosphate from d-fructose 6-phosphate (28). The lma operon transcript was detected only at low temperatures, and the mutant Δ*lmaD* showed attenuated virulence compared with the wild type (29, 30). Mutants with an insertion in the rhamnose transport and utilization operon (*lmo2846-lmo2851*) showed attenuated growth at low temperatures. Previous studies showed that rhamnose was used as a carbon source, a decoration of the cell wall teichoic acids, and required for adsorption of A118 like bacteriophages in *Listeria monocytogenes* (31, 32). The lma operon was essential for the growth of *Listeria monocytogenes* at low temperatures.

Two-component systems (TCSs), consisting of a membrane-bound histidine kinase and a response regulator, may aid bacterial adaptation to various stressful situations. In *Clostridium botulinum*, the two-component system Clo3403/Clo3404 was critical for cold tolerance (33). However, no clear contributions of two-component regulatory systems of Lm were identified by our Tn-seq results (Fig. 3).

Notably, mutant insertions in A118 phage-associated genes could survive better at 16°C than at 37°C (Fig. 3). A118 is a temperate phage of Lm and can be used as a biocontrol for Lm (34, 35). A previous study demonstrated that the plaquing and adsorption efficiencies of Lm phages were significantly affected by temperature (36).

All these findings validated the effectiveness of the Tn-seq approach. As shown in Fig. 3, the genes (*cspL*, *bipA*, *pfkA*, *lmaB*, *rhlA*, *rhlD*) associated with low temperatures had been reported previously. We focused on the genes that had not been reported yet with significant fold change (|FoldChange|≥2, BH ≤ 0.01). As a result, we chose *lmo1366* and *lmo2301* as prospective genes for further study.

A

```
Lmo1366   MTIKKERADILLVEQGLFETREKAKRAIMAGIVYRKEERVDKPGEKIPIDSELQVKGKQM      60
YqxC      MTSKKERLDVLLVERGLAETREKAKRAIMAGIVYSNENRLDKPGEKIDRDLPLTVKGNPL
TylA          MRLDEYVHSEGYTESRSKAQDIILAGCVFVNGVKVTSKAHKIKDTDNIEVV-QNI

Lmo1366   PYVSRGGLKLEKALQVFDFEVKDKLMLDIGASTGGFTDCALQNGARHSYALDVGYNQLAW     120
YqxC      RYVSRGGLKLEKALKEFPVSVKDKIMIDIGSSTGGFTDCALQNGAKQSYAVDVGYNQLAW
TylA      KYVSRAGEKLEKAFVEFGISVENKICLDIGASTGGFTDCLLKHGAKKVYALDVGHNQLVY

Lmo1366   KLRNDDRVTVMERTNFRHVKPADFAEGLADFATIDVSFISLKLILPVLRTVLVTGGDVMT     180
YqxC      KLRQDERVVVMERTNFRYATPADFTKGMPEFATIDVSFISLRLILPVLRTLLVPGSDCMA
TylA      KLRNDNRVVSIEDFNAKDINKEMFNDEIPSVIVSDVSFISITKIAPIIFKELNNLEFWVT

Lmo1366   LLKPQFEAGREQVGKKGIIRDPAVHESVVEHIVQFALDNGYDLMGLDYSPITGGEG        240
YqxC      LVKPQFEAGRESVGKKGIVRDPKVHADVLKRMISFSAAEGYICKGLSFSPITGGDGNIEF
TylA      LLKPQFEAERGDVSKGGIIRDDILREKILNNAISKIIDCGFKEVNRTISPIKGAKGNIEY

Lmo1366   IAHLKWTGEETGISHLEPDAIAKLITKAHTKLDK
YqxC      LLHLHWPGEGQEGQELPEEEIMRVVEEAHKTLKEKKADVPE
TylA      LAHFII
```

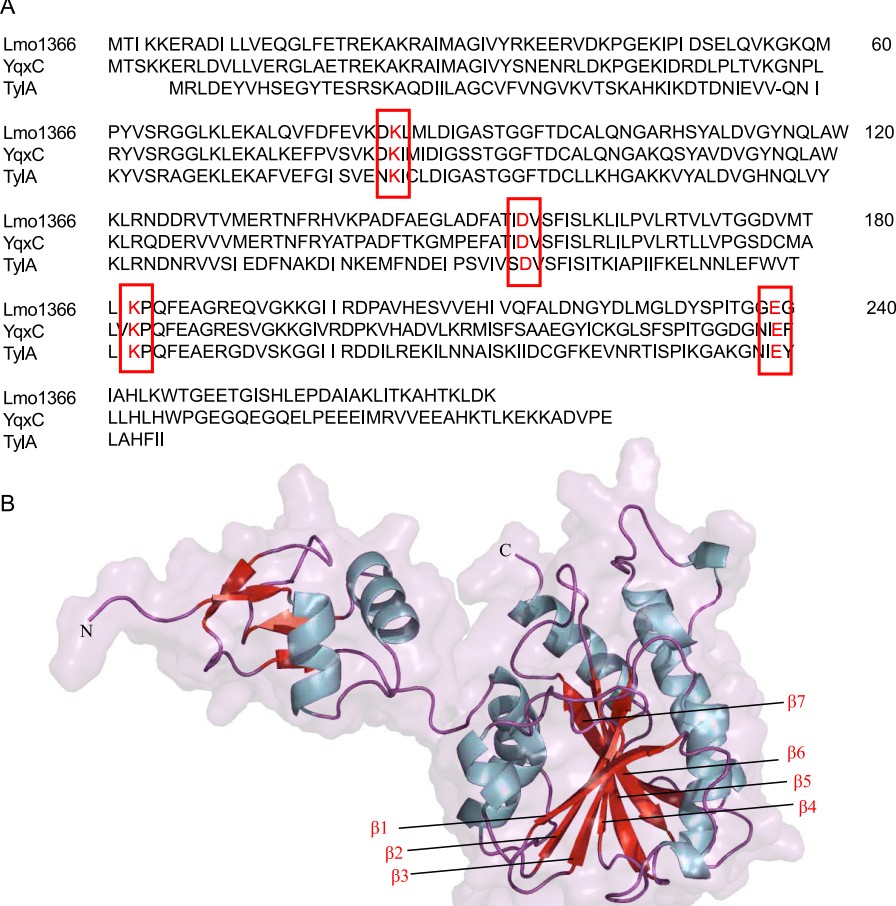

**FIG 4** Amino acid sequence and protein structure of Lmo1366. (A) Amino acid sequence alignment of Lmo1366 from *L.monocytogens*, YqxC from *Bacillus subtilis*(strain 168), and TylA from *Mycobacterium tuberculosis* (ATCC 25618). The catalytic tetrad was shown in boxes. (B) The diagram of the Lmo1366 protein tertiary fold. Lmo1366 comprises an N-terminal ribosomal protein S4-fold and a C-terminal class I methyltransferase fold with a seven $\beta$-strand core ($\beta 1-\beta 7$) surrounded by $\alpha$-helices.

**In silico analysis of the Lmo1366 and Lmo2301 protein sequences.** Bioinformatics studies revealed that the Lmo1366 protein contains motifs aligned well with an rRNA methyltransferase. This enzyme was composed of two protein domains: S4 and FtsJ. The S4 domain starting at the N terminus consisted of 62 amino acids. The S4 domain was probably responsible for the initial interaction with the RNA or a complex of RNA and ribosomal proteins. The S4 domain allowed the formation of a stable enzyme-substrate complex and carried out the methylation reaction. The second domain, FtsJ, contains four residues: K84-D156-K183-E249, corresponding to the K-D-K-E tetrad in *E. coli* RrmJ (FtsJ) methyltransferase responsible for 2′-O methylation of U2552 in the A loop of 23S rRNA (Fig. 4). The FtsJ domain was responsible for the methylation of ribonucleic acids using S-adenosylmethionine as the methyl group donor (37).

*In silico* three-dimensional modeling of the Lmo1366 protein showed a structure typical of RNA methyltransferases (Fig. 4). The smaller S4 domain covered the enzyme's active site, located in the middle of the FtsJ domain. A characteristic feature is the arrangement of $\alpha$-helices and $\beta$-sheets in the FtsJ domain, constituted by 7 $\beta$-sheets surrounded by 7 $\alpha$-helices – a layout very typical for RNA methyltransferases. A similar arrangement occurs in the larger domain of the TlyA protein of *Mycobacterium tuberculosis* (38) and YqxC of *Bacillus subtilis* (39).

The *lmo2301* gene encodes the terminase of phage A118 of Lm. A118 is a temperate phage, and the entire nucleotide sequence and structural analysis had been

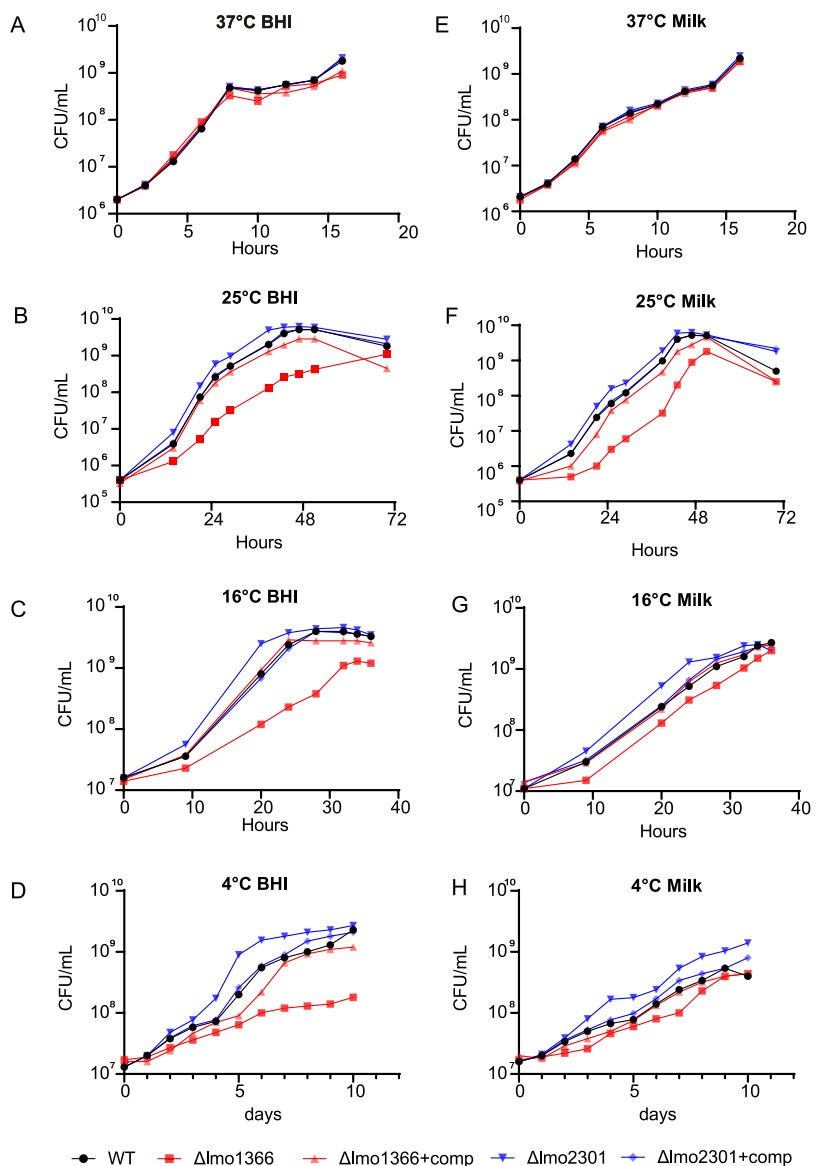

**FIG 5** Growth curves of the WT (●), Δ*lmo1366* (▲), Δ*lmo1366*+comp (■), Δ*lmo2301* (▼), Δ*lmo2301*+comp (◆) in BHI (A–D) or milk (E–H) at different temperatures (37°C, 25°C, 16°C, 4°C).

reported previously (34). A study revealed that A118 modulates virulence as a genetic switch promoting phagosomal escape (24).

**Growth profiles of the mutant strains.** The deletion mutants and their complements were successfully constructed, and then their phenotypes were analyzed. In order to test more wide-range low-temperature conditions, the mutant strains were grown at 4°C, 16°C, 25°C, and 37°C in BHI. The results showed no significant differences between the two mutants and wild-type strain at 37°C (Fig. 5A). At low temperatures, however, Δ*lmo1366* grew slower than the wild-type strain, and the mutant strains' growth phenotype was restored in the complemented strain (Fig. 5B to D, red). This result was also consistent with the Tn-seq data. Our results confirmed that *lmo1366* was an essential gene at low temperatures.

Moreover, a similar slow growth trend was observed in milk at low temperatures (Fig. 5E to H, red). Milk contains various macronutrients and micronutrients, including vitamin B12, vitamin A and calcium (40). Moreover, multiple listeriosis outbreaks were retrospectively associated with contaminated milk (41). These results could help us better understand the possible molecular mechanisms of Lm survival in milk at low temperatures.

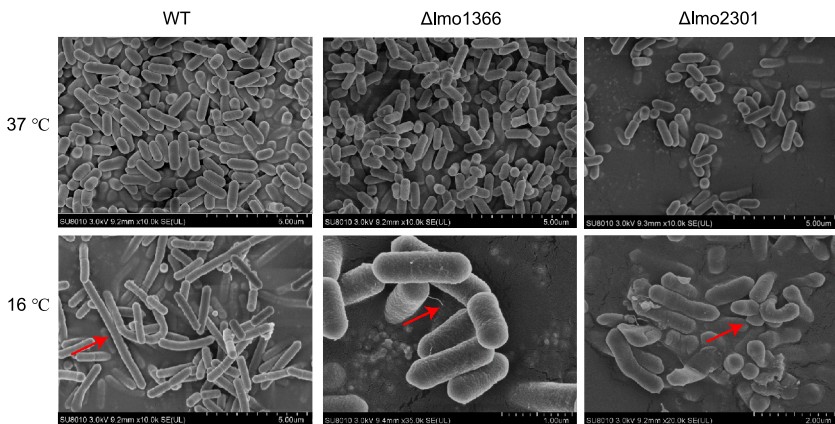

**FIG 6** Scanning electron microscope (SEM) images of WT, Δ*lmo1366*, and Δ*lmo2301* at 37℃/16℃. The red arrows pointed at the striking changes. WT at 16℃ became more elongated than 37℃. Δ*lmo1366* at 16℃ had cell membrane roughness and hair-like pilus structures protruding from the cell surfaces. Δ*lmo2301* at 16℃ could not maintain the normal morphology.

The Tn-seq results showed that insertions in *lmo2301* resulted in a higher fitness at 16℃ (11-fold). We also observed that the mutant strain Δ*lmo2301* grew more rapidly, and this growth phenotype was partially restored in the complemented strain (Fig. 5B to D, blue). However, the fold change was much smaller than the Tn-seq results. These results confirmed that *lmo2301* is a negative-regulate gene at low temperatures.

To examine the heat tolerance of Δ*lmo1366* and Δ*lmo2301*, we performed the bacterial colony counting at 62℃ after 30 min in BHI or milk. The results showed no significant differences between the wild-type strain and the mutants (Fig. S1), suggesting that *lmo1366* and *lmo2301* do not exert a unique role at high temperatures.

To determine whether *lmo1366* and *lmo2301* are associated with other factors, we performed the growth curve of mutants at different pH and osmotic pressure conditions. The results showed no significant difference between mutants and wild type, indicating that *lmo1366* and *lmo2301* are specifically associated with low temperatures (Fig. S4, Fig. 5E).

Although the targeted mutant strains showed consistent results at 4℃ and 16℃, it did not represent that all the genes screened by Tn-seq at 16℃ would necessarily get consistent results at 4℃.

**Morphological changes of the mutant strains.** Scanning electron microscopy (SEM) analysis of bacteria was carried out to observe morphological changes. Smooth and intact short rods of wild-type strains were observed at 37℃, similar to previous findings (42). There were no apparent differences in morphology between the two mutants and the wild-type strain at 37℃ (Fig. 6). The wild-type strains displayed a striking change in cell shape, becoming more elongated at 16℃ (Fig. 6). The cell elongation response to stress agreed with previous investigations (43, 44).

For Δ*lmo1366*, we observed cell membrane roughness and hair-like pilus structures protruding from the cell surfaces at 16℃. Moreover, the mutant Δ*lmo2301* could not maintain the normal morphology at 16℃ (Fig. 6). The morphology results could explain the fold change difference in the Tn-seq results and the growth curve of Δ*lmo2301*. Tn-seq results showed that the fold change of Δ*lmo2301* at 16℃ versus 37℃ was 11 times (Fig. 3), while the growth curve of Δ*lmo2301* indicated the fold change was much smaller than 11 times (Fig. 5C). Since Tn-seq could sequence the mutants with abnormal morphology, they could not form colonies of typical size at CFU counting. Overall, the morphological changes of the mutant strains Δ*lmo1366* and Δ*lmo2301* were associated with growth at low temperatures.

**Transcriptome analysis of mutants at 16℃.** To explore the possible mechanisms by which *lmo1366* affects the growth of Lm at low temperatures, RNA-seq was performed. The results showed that 392 genes were upregulated, and 409 were downregulated significantly (|log2FC|≥1, padj ≤ 0.01). The specific differentially expressed

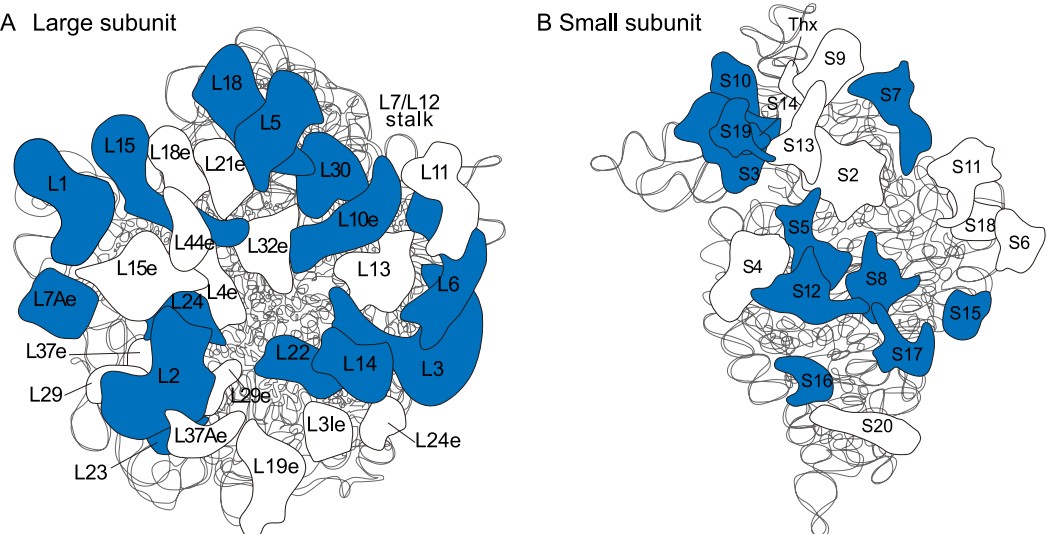

**FIG 7** Analysis of the differentially expressed genes. (A) Large unit of 50S ribosomal proteins. (B) Small unit of 30S ribosomal proteins. The label presented the ribosomal protein name. The blue presented the downregulation.

genes are listed in Table S2. KEGG analysis revealed that genes downregulated in Δ*lmo1366* at 16°C were significantly enriched among flagellar assembly and ribosome genes (Fig. S2A). GO analysis revealed that genes upregulated in Δ*lmo1366* at 16°C were significantly enriched among nucleotide-binding and carboxylic acid metabolic processes (Fig. S2B).

Exposure to low temperatures can affect ribosomal structural stability in the bacterium (45). The 50S ribosomal proteins were the first cold stress sensors in microbes. Previous transcriptomic analysis of Lm showed higher transcript levels of ribosomal protein genes at 4°C compared to wild type at 37°C (46). Moreover, shotgun proteomics analysis showed more ribosomal proteins at 25°C than 37°C (47). The transcriptional upregulation of ribosomal protein genes helped in protein folding, protection against thermal denaturation, and interaction with unfolded proteins at low temperatures of Lm (48). However, our results showed that transcription of the ribosomal protein genes decreased significantly in Δ*lmo1366* versus wild type at 16°C (Fig. 7). Our findings suggested that *lmo1366* encoding rRNA methyltransferase severely compromises ribosome structural stability and function, leading to mutant growth defects at low temperatures.

Previous studies indicated that the protein folding machinery that causes protein denaturation and aggregation is crucial for survival under environmental stress conditions (49). Chaperone proteins associated with the general stress response system, such as DnaK, DnaJ, GroES, GroEL, and GrpE, are less abundant at low temperatures (47). However, our results showed that the molecular chaperone genes *dnaK* and *lmo0292* displayed more significant fold changes than wild type at 16°C (Table S2).

Transcriptomic analysis of Δ*lmo2301* growth at 16°C showed that 379 genes were upregulated, and 312 genes were downregulated significantly ($|log2FC|≥1$, padj$<0.01$) (Table S2). KEGG analysis revealed that genes downregulated of Δ*lmo2301* at 16°C were significantly enriched in the phosphotransferase system (PTS) and fructose metabolism (Fig. S2C). Furthermore, genes upregulated in Δ*lmo2301* at 16°C were significantly enriched in peptidoglycan biosynthesis and fatty acid metabolism (Fig. S2D). A previous study showed that the upregulation of genes involved in membrane biogenesis, such as fatty acid and LPS biosynthesis, peptidoglycan biosynthesis, glycosyltransferases, and outer membrane proteins, helped the survival of the bacterium at low temperatures (50, 51). Our results suggested that *lmo2301*, encoding the terminase of phage A118, was associated with membrane biogenesis, leading to the enhanced growth of the mutant at low temperatures.

To confirm the RNA-Seq results, we analyzed the transcription levels of nine genes by qPCR (Fig. S3). The graphic shows the consistency between the RNA-seq results and RT-qPCR results ($R^2$ = 0.9985). Accordingly, the reliability and accuracy of the RNA-seq data were confirmed.

**Conclusion.** Understanding the mechanism of Lm growth at low temperatures is important to find practical approaches to restrict Lm growth in food control. In this study, we applied a powerful functional genomics method Tn-seq to identify novel genes involved in the low-temperature growth of Lm. One hundred and 40 genes were identified, including many genes contributing to low-temperature growth that have been reported in previous studies, which proved the validity of our results. Furthermore, we looked deeper into two novel genes involved in growth under low-temperature conditions. The results showed the growth defects of Δ*lmo1366* and the enhanced growth of Δ*lmo2301*. Transcriptome analysis suggested that the rRNA methyltransferase gene *lmo1366* played an essential role at low temperatures by affecting ribosome structural stability and function. In contrast, the terminase of phage A118 gene *lmo2301* played a negative role in growth at low temperatures by affecting membrane biogenesis. The deletion of *lmo1366* or *lmo2301* resulted in cell morphology alterations at 16℃ by affecting membrane biogenesis.

Collectively, these results reveal novel mechanisms of low-temperature growth in *L. monocytogenes* and may provide unique antimicrobial targets for the development of novel strategies to control foodborne pathogens in low-temperature storage or processing environments. Targeting rRNA methyltransferases can inhibit translation and eliminate the ability to methylate its substrates hence inhibiting the growth of Lm at low temperatures. In the future, applying theoretical results to food control through scalable approaches is the focus of our research, and more related research is warranted.

## MATERIALS AND METHODS

**Strains, media, and reagents.** The bacterial Lm EGD-e (ATCC BAA-679) strain was obtained from American Tissue Culture Collection. The strains were grown in brain heart infusion (BHI) broth or BHI agar at 37℃ or 16℃. Pasteurization processed milk was purchased from a local grocery store. Antibiotics (Solarbio) were used at the following concentrations: chloramphenicol 10 $\mu$g/mL, gentamicin 25 $\mu$g/mL, penicillin 50 $\mu$g/mL, nisin 25 ng/mL, spectinomycin 100 $\mu$g/mL. erythromycin 50 $\mu$g/mL, lincomycin 50 $\mu$g/mL.

**Generation of the mutant library.** The high-density transposon mutant library was previously constructed by our lab, which had 63666 unique insertion mutants. The details were described previously (52). The 10 $\mu$L plasmid pGPA-2 was transferred into the 100 $\mu$L competent Lm by electrotransformation. Transformants on the BHI plate (with chloramphenicol and gentamicin) were grown overnight in BHI broth (with chloramphenicol) at 30℃, then 200 $\mu$L of this overnight culture was added to 5 mL of prewarmed BHI broth (with nisin and gentamicin) and grown overnight at 30℃. Then 200 $\mu$L of overnight culture was added to 100 mL of prewarmed BHI broth and grown overnight at 37℃. Subsequently, 2 mL of overnight culture was added to 100 mL of prewarmed BHI broth (with gentamicin) and grown overnight at 37℃. Cultures were stored in BHI broth containing 50% (vol/vol) glycerol at −80℃ as the mutant library.

**DNA library preparation for Tn-seq sequencing.** In order to prepare the DNA for Tn-seq sequencing, 1 mL transposon mutant library bacterial cultures growth at 16℃and 37℃ were centrifuged, and the genomic DNA was extracted. Two micrograms of the extracted DNA were digested for 30 min at 37℃ using 10 U MmeI (New England Biolabs) and immediately dephosphorylated with 1 U of calf intestine alkaline phosphatase (Invitrogen) for 60 min at 50℃. The dephosphorylated DNA was purified using a SanPrep PCR purification kit (Sangon Biotech) and dissolved in 30 $\mu$L ddH2O. Subsequently, Tn-seq phosphorylated adapters with a 6-bp barcode were prepared. Next, ligation of 100 ng dephosphorylated MmeI restriction fragments with two pmol phosphorylated adapters was performed using 2 U T4 DNA ligase (New England BioLabs) for 15 min at room temperature. Immediately after the ligation, 2.5 $\mu$L ligation reaction product and 20 pmol primers were PCR using the Phusion High-Fidelity PCR Master Mix with HF Buffer kit (Thermo Scientific). The PCR cycling conditions were as follows: 72℃ for 1 min and 98℃ for 30 s; 25 cycles of 98℃ for 30 s, 57℃ for 30 s, and 72℃ for 10 s; and 72℃ for 5 min. The PCR products of 128 bp were purified using the Minelute reaction cleanup kit (Qiagen). Then all the purified products were mixed at the same quantity (ng) according to their concentration measured by nanodrop. Tn-seq samples were sequenced (HiSeq PE150) on one lane of an Illumina HiSeq X 10 (Shanghai Personal Biotechnology Co.). The experiment was performed with three biologically independent replicates.

**Tn-seq data analysis.** ARTIST analysis was performed to identify essential genes involved in cold temperature resistance. The raw data were demultiplexed and trimmed using Fastp (53). Reads with 16 bp were mapped to the Lm genome using bowtie2 (54). The scripts were used to identify reads corresponding to insertions in TA sites and tally only those insertions with reads mapping from both ends of the transposon (55). Genes were screened according to fold change and BH value (|FC|≥2, BH ≤ 0.01).

**Protein prediction.** The protein sequence was uploaded to alphafold2.1 software for protein domain prediction. Five pdp files output by alphafold were obtained, and PyMOL (Version 2.0.7) software was used to open the pdp files for analysis.

**Mutants and the complements construction.** To construct Δ*lmo1366* and Δ*lmo2301*, we used the Cre-*loxP* recombination system. The details were described previously (56). Primers were listed in Table S3. SmaI digested the plasmid pWS3, and the PCR segments (gene_up, gene_dn, gm) were ligated by NovoRec plus One step PCR Cloning kit. The plasmid pWS3_gene_gm was transferred into the competent Lm by electrotransformation. Transformants on the BHI plate (with gentamicin and spectinomycin) were grown overnight in BHI broth at 37°C. Δgene::gm colonies that only grow on plates (with gentamicin) but not on plates with spectinomycin were confirmed by PCR using the check primers.

The 10 $\mu$L plasmid pRAB1-erm was transferred into the 100 $\mu$L competent Δgene::gm by electrotransformation. Transformants on the BHI plate (with erythromycin and lincomycin) were grown overnight in BHI broth at 37°C. Δgene colonies that only grow on BHI plates but not on plates with gentamicin were confirmed by PCR using the check primers.

The plasmid pMSP3535 was digested by pstI, and the PCR segments (comp_gene) were ligated. The plasmid pMSP3535_comp_gene was transferred into the competent DH5$\alpha$ by heat shock transformation. Transformants on the BHI plate (with 100 $\mu$g/mL erythromycin) were grown overnight in BHI broth at 37°C. The recombinant plasmid was transferred into the competent Δgene by electrotransformation. Δgene::comp transformants on BHI plate (with erythromycin and lincomycin) were grown overnight in BHI broth at 37°C and confirmed by PCR with comp_gene_F/R primers.

**Morphological observation of Lm.** Lm was cultured in BHI at 16°C and 37°C until the exponential phase. Then the bacteria were observed by SEM (Hitachi SU-8010) in the Bio-ultrastructure Analysis Lab of the Analysis Center of Agrobiology and Environmental Science (Zhejiang University).

**RNA-seq.** Lm was cultured in BHI at 16°C and 37°C until the exponential phase. The bacterial solutions were centrifuged (5 min; 4000 rpm) and quickly frozen in liquid $N_2$. After assessing the total amounts and integrity of RNA, the mRNA was obtained by removing rRNA (rRNA). Then, a fragment buffer was added to break the obtained mRNA into short fragments randomly, and the library was built by the chain-specific library. The RNA extraction, library preparation, and transcriptome sequencing were completed by Novogene (Beijing).

**RT–qPCR.** RNA extraction was performed as described above. RT–qPCR was performed with the following cycling protocol: 1 cycle at 95°C for 3 min;40 cycles at 95°C for 10 s, 56°C for 20 s, and 72°C for 20 s. Primers were listed in Table S3. Three parallel experiments were performed for each gene detected by RT–qPCR (*argC*, *pyrP*, *nadA*, *gbuB*, *lmo2097*, *motB*, *rplB*, *flaA*) with three technical replicates. Data were normalized to a housekeeping gene and analyzed by the comparative threshold method (△△Ct).

**Data availability.** Illumina sequencing reads (Accession No. SRR18767871-SRR18767875) and RNA-seq (Accession No. SRR18769210-SRR18769218) have been submitted to the NCBI server.

## SUPPLEMENTAL MATERIAL

Supplemental material is available online only.
**SUPPLEMENTAL FILE 1**, PDF file, 0.7 MB.
**SUPPLEMENTAL FILE 2**, XLSX file, 2.1 MB.

## ACKNOWLEDGMENTS

This study was supported by the Project of Shandong Province Higher Educational Outstanding Youth Innovation Team [2019KJF011], the Natural Science Foundation of Shandong Province, China [ZR2019ZD21], the Taishan Scholars Program of Shandong Province, China [ts20190955], and the National Key Research and Development Program of China [2019YFE0103900].

We declare that we have no known competing financial interests or personal relationships that could have appeared to influence the work reported in this paper.

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
