## [Reviewer comments · Microbiology Spectrum]

Microbiology Spectrum

Functional genomics identified novel genes involved in growth at low temperatures in *Listeria monocytogenes*

Yansha Wu, Xinxin Pang, Xiayu Liu, Yajing Wu, and Xinglin Zhang

Corresponding Author(s): Xinglin Zhang, Zhejiang University

Review Timeline:

Submission Date:	February 24, 2022
Editorial Decision:	April 11, 2022
Revision Received:	April 28, 2022
Editorial Decision:	May 28, 2022
Revision Received:	June 2, 2022
Accepted:	June 4, 2022

Editor: Luca Cocolin

Reviewer(s): Disclosure of reviewer identity is with reference to reviewer comments included in decision letter(s). The following individuals involved in review of your submission have agreed to reveal their identity: Augustine Kwaku Agyekum (Reviewer #2)

Transaction Report:

DOI: <https://doi.org/10.1128/spectrum.00710-22>

April 11, 2022

Prof. Xinglin Zhang
Zhejiang University
Yuhangtang Rd.866
hangzhou
China

Re: Spectrum00710-22 (Functional genomics identified novel genes involved in growth at low temperatures in *Listeria monocytogenes*)

Dear Prof. Xinglin Zhang:

Link Not Available

Sincerely,

Luca Cocolin

Journals Department
Reviewer comments:

Reviewer #3 (Public repository details (Required)):

They have done Tn-seq and RNA-seq, however they do not mention if these have been deposited with any repository.

Reviewer #3 (Comments for the Author):

See attached file.

Reviewer #4 (Comments for the Author):

Wu et al. have used functional genomic analyses via Tn-seq to identify novel genes in *L. monocytogenes* involved in growth at low temperatures. The study methods, results and conclusions are sound and the findings of novel genes involved in bacterial growth at low temperature are notable. The paper would be greatly strengthened, however, by addressing the following:

Major:

1. The authors should describe Tn-seq principle in more detail in introduction, as the overall principle is not clear.
2. In the results, it is still unclear why exactly lmo1336 and lmo2301 were selected for further study. Please clarify why these looked to be among the most promising markers based on the data presented (ln 145).
3. The authors focus on temperature regulation and thermal survival pathways, but it's unclear whether these same pathways are truly specific to temperature or if pH, nutrients, etc. also play a role. It would be helpful to also look at morphology in milk cultures, for example. Is any of this data available?
4. The authors suggest that findings could contribute to development of novel antibacterial strategies but don't elaborate on this in sufficient detail. How, exactly, could these findings translate to *Lm* sterilization or prevention strategies and what should be the priorities for future research and implementation efforts?

Minor:

Italicize genes throughout

Please review throughout for punctuation and spacing errors in addition to language throughout, with special attention to past and present tense for consistency

Ln49-50 Combine sentences

Ln54: delete word "invasive"

Ln 55: change to "immunosuppressed"

Ln 62: "novel food vehicles"- please explain or replace

Ln 73: Sentence doesn't necessarily follow. Instead say that knowledge is incomplete

Ln 166: Change to "complements"

Fig 3: Would suggest making background circles an even lighter color to reduce noise

Fig 4: Make it clear how image was generated.

Fig S1: Were any statistics applied here? Any significant findings?

Staff Comments:

Preparing Revision Guidelines

Please return the manuscript within 60 days; if you cannot complete the modification within this time period, please contact me. If you do not wish to modify the manuscript and prefer to submit it to another journal, please notify me of your decision immediately so that the manuscript may be formally withdrawn from consideration by Microbiology Spectrum.

Yansha Wua et al.

Functional genomics identified novel genes involved in growth at low temperatures in *Listeria monocytogenes*.

The manuscript describes work in which the authors identify two genes with possible roles in growth of *Listeria monocytogenes* at low temperatures. They maintain that lmo1366 plays an essential role in the growth of Lm at 16°C, while lmo2301 significantly enhanced the growth of Lm at 16 °C. The deletion of lmo1366 or lmo2301 resulted in cell morphology alterations and impaired growth rate in milk and in BHI at low temperatures.

Although the design of the study has no flaws, and the results are well documented and interpretation is acceptable, I have two major concerns regarding this work.

1. There is no justification of why the authors have chosen to analyze the growth characteristics of Lm at 16°C. My opinion is that 16°C is not a preferred storage temperature for foods in which Lm is likely to grow and cause food contamination e.g., milk. Although they have one experiment on growth kinetics of mutant and complementation strains at 4°C, they originally generated the mutants at 16°C. Would they have similar conclusions had they generated the mutants at 4°C? In fact, they do not discuss the problem of food contamination by Lm at all, neither in the introduction nor Results/Discussion section. Besides, most of their references for low temperature Lm growth studies refer to temperatures well below 16°C.

2. The methodology is not detailed. For example (1) transposon mutant generation is not described but only a general reference is provided, (2) Cre-lox recombination system used is not described.

Other concerns

1. English language still requires some work. Authors should consult a native English-speaking colleague or an English editing service.

2. How relevant are the mutations that allow growth at 16°C? Storage temperature for food is 4-8°C. The study does not provide the rationale for choosing 16°C as a low temp for Lm growth.

3. Line 61-64 - There is need to understand the mechanisms of growth at low temperature, but definitely not a great need since only 123 cases were registered over a period of 2 decades.

4. Line 91-92 - where is this previous study. was it published? If so, please cite.

5. Line 104 – “Like” is not necessary here. Correct.

6. Line 133 - This should be clarified that the requirement for greater rhamnose concerns *E. coli* and not Lm, at least that is what the reference is about.

7. Line 158 - How does the reader visualize this?

8. Line 197-199 - please rephrase this sentence. It does not render the meaning intended.

9. Line 256 - "or" - please correct.

10. Line 257 - You could provide a lead statement of how this could be possible.

11. Although not the aim of this study, the authors argue in the conclusion that finding mechanisms of Lm low temp growth could help find effective approaches to limit Lm growth in foods stored at lower temps. However, they fail to discuss how the obtained results could map a way forward in that direction.
12. Line 266-269. The reference provided for transposon mutant generation is a general approach to such work. What was the specific protocol designed for the work described in the manuscript?
13. Line 297-299 - Understandably, a powerful gene expression control system with many modifications to custom the gene control required. The authors have not informed the reader which approach of this system they used, neither do they cite work where the approach could have been used. Similarly, the complementation system, did the authors use it exactly as reported in the cited paper or did they adapt it to perform their work. Those details are lacking for others to be able to replicate these results should there be need to do so.
14. Line 306-308. Perhaps this line should come last in this paragraph.
15. Line 315-316 -what groups are the authors referring to?
16. Line 458- labels 1, 2, 3 and 4 are missing on the figure.
17. Line 464-465 -which X-axis are the authors referring to in Figure 1?
18. Fig 3 - What analytical software was used to generate the data plotted in this figure?
19. Line 473 - these are not circles. they are spheres.
20. Line 474 - But why is the scale only up to -6 and +8, if change fold was e.g., 13 for lmo1366?

Response to reviewers' comments

Dear Luca Cocolin,

On behalf of my co-authors, we thank you very much for giving us an opportunity to revise our manuscript, we appreciate editor and reviewers very much for their positive and constructive comments and suggestions on our manuscript entitled “Functional genomics identified novel genes involved in growth at low temperatures in *Listeria monocytogenes*”.

We have studied the comments of editor and reviewers carefully and have made revision which marked in red in the revised manuscript. We have tried our best to revise this manuscript according to the comments. Attached please find the revised version, which we would like to submit for your kind consideration. And responses to reviewers' comments are appended below.

We would like to express our great appreciation to you and reviewers for comments on our paper. Looking forward to hearing from you.

Yours sincerely,

Dr. Xinglin Zhang
Department of Food Science and Nutrition
Zhejiang University
Hangzhou 310058 P.R.China
Tel: +86-571-86984316
E-Mail: zhangxinglin@lyu.edu.cn

List of Responses.

Reviewer #3:

1. The manuscript describes work in which the authors identify two genes with possible roles in growth of *Listeria monocytogenes* at low temperatures. They maintain that lmo1366 plays an essential role in the growth of Lm at 16°C while lmo2301 significantly enhanced the growth of Lm at 16°C. The deletion of lmo1366 or lmo2301 resulted in cell morphology alterations and impaired growth rate in milk and in BHI at low temperatures. Although the design of the study has no flaws, and the results are well documented and interpretation is acceptable, I have two major concerns regarding this work.

Reply: We thank the reviewer for his/her compliments on our manuscript. Below we will reply in a point-by-point fashion to the reviewer's comments and indicate the alterations we have made to the manuscript.

2. There is no justification of why the authors have chosen to analyze the growth characteristics of Lm at 16°C. My opinion is that 16°C is not a preferred storage temperature for foods in which Lm is likely to grow and cause food contamination e.g., milk. Although they have one experiment on growth kinetics of mutant and complementation strains at 4°C, they originally generated the mutants at 16°C. Besides, most of their references for low temperature Lm growth studies refer to temperatures well below 16°C.

Reply: We thank the reviewer for their constructive point. The reason why we choose 16°C rather than 4°C were followings:

a) We agree that 4°C is closer to the temperature of actual refrigeration, however we have to strike a balance between the optimal experimental condition and the technical feasibility. The preliminary results showed that it took 3 weeks for *L. monocytogenes* to grow from 10⁶ to 10⁹ CFU at 4°C, however, only took 2 days at 16°C. Because Tn-seq is very sensitive, the 4°C condition could impose too much stress for screening, which would lead to over-selection and decreased specificity.

b) The purpose of this experiment is to identify genes that play an important role in growth under low temperature conditions. As the reviewer has noted, the genes identified at 16°C were further characterized at 4°C and other conditions.

c) Previous studies also used 16°C as the low temperature research condition in *L. monocytogenes*. Patrick et al selected conditions that reflect temperatures that may be encountered by *L. monocytogenes* in the environment and in cold-blooded animals (16°C and 30°C) or warm-blooded hosts (37°C and 42°C)(1). Oluwadara et al assessed the survival behavior of *L. monocytogenes* in 14 different types of ready-to-eat vegetable salads stored at 4, 8, 12 and 16 °C(2). Grzegorz et al assessed the survival of *L. monocytogenes* in four low moisture foods stored at 16 °C(3). Xanthiakos et al monitored the growth of *L. monocytogenes* in a commercial pasteurized whole milk product at 16°C(4). The inoculated ham samples were stored at 4, 8, and 16 °C, and *L. monocytogenes* was enumerated periodically during the storage(5). M.Gougoul et al assessed the kinetic behavior of *L. monocytogenes* in ice cream stored under static chilling conditions (4 to 16°C)(6).

d)To sum up, we choose 16°C as the research condition.

3. Would they have similar conclusions had they generated the mutants at 4°C?

Reply: Thanks for reviewer's question. We would expect more (hundreds of) genes were identified contributing to low-temperature growth if the Tn-seq screening were performed at 4°C. However, as we replied above, these genes may not be specifically contributing to low-temperature growth, due to the over-loaded stress

and the too-slow growth rate. Although the Tn-seq screening at 16°C may miss some genes, we think the most important and specific genes that contributing to low-temperature growth will be identified.

4. In fact, they do not discuss the problem of food contamination by Lm at all, neither in the introduction nor Results/Discussion section.

Reply: We appreciate for reviewer's advice. We only discussed the problem of food contamination by Lm in general terms in the introduction section (Line64-67). Following the reviewer's suggestion, we expanded it into more detail. (Line67-71).

5. The methodology is not detailed. For example (1) transposon mutant generation is not described but only a general reference is provided, (2) Cre-lox recombination system used is not described.

Reply: We thank the reviewer for their constructive point. Following the reviewer's suggestion, we have described the methodology in detail in the revised version. (Line303-314, Line349-367)

6. English language still requires some work. Authors should consult a native English-speaking colleague or an English editing service.

Reply: We appreciate for reviewer's advice. Following the reviewer's suggestion, we have revised the manuscript to improve the English language by American Journal Experts, the English editing service recommended by ASM.

7. How relevant are the mutations that allow growth at 16°C? Storage temperature for food is 4-8°C The study does not provide the rationale for choosing 16°C as a low temp for Lm growth

Reply: We thank the reviewer for their constructive point. As we stated above in Reply 2 and 3, previous studies also used 16°C as the low temperature research condition in *Listeria monocytogenes*(2-6) . Following the reviewer's suggestion, we

provide the rationale for choosing 16°C as a low temp for Lm growth in the introduction section (Line67-71).

We have to strike a balance between the optimal experimental condition and the technical feasibility. Tn-seq is a very sensitive and tricky technique, the 4°C condition could impose too much stress for screening, which would lead to over-selection and decreased specificity. Over-loaded stress condition may lead to identification of hundreds of genes, in which most genes may not have strong or specific correlation to low-temporization growth.

8. Line 61-64 - There is need to understand the mechanisms of growth at low temperature, but definitely not a great need since only 123 cases were registered over a period of 2 decades.

Reply: We appreciate for reviewer's advice and we have rewritten this sentence in a more proper way. The 123 cases only in ProMED database, and there were more cases reported in other databases. Although the number is not huge, the mortality rate was really high(8). (Line62-63)

9. Line 91-92 - where is this previous study. was it published? If so, please cite.

Reply: Thanks for reviewer's question. The study is not published yet. It's in submission. Following the reviewer's suggestion, unpublished data are annotated in the article (Line94)

10. Line 104 – "Like" is not necessary here. Correct.

Reply: We appreciate for reviewer's advice and we have corrected it. (Line 118)

11. Line 133 - This should be clarified that the requirement for greater rhamnose concerns E. coli and not Lm, at least that is what the reference is about.

Reply: We appreciate for reviewer's advice and we have corrected it. We revised it by citing the reference of *L. monocytogenes* to illustrate the role of lma operon in *Listeria*. (line149-152)

12. Line 158 - How does the reader visualize this?

Reply: We appreciate for reviewer's constructive question. We have re-drawn the Fig.4B and added descriptions to make it easier for readers to understand. (Line589-591)

13. Line 197-199 - please rephrase this sentence. It does not render the meaning intended.

Reply: We appreciate for reviewer's advice and we have rephrased this sentence. (Line222-226)

14. Line 256 - "or" - please correct.

Reply: We appreciate for reviewer's advice and we have corrected it. (Line286)

15. Line 257 - You could provide a lead statement of how this could be possible.

Reply: We appreciate for reviewer's advice and we have rephrased this sentence. (Line287)

16. Although not the aim of this study, the authors argue in the conclusion that finding mechanisms of Lm low temp growth could help find effective approaches to limit Lm growth in foods stored at lower temps. However, they fail to discuss how the obtained results could map a way forward in that direction.

Reply: We appreciate for reviewer's advice and we added it in conclusion. (Line290-293)

17. Line 266-269. The reference provided for transposon mutant generation is a general approach to such work. What was the specific protocol designed for the work described in the manuscript?

Reply: We appreciate for reviewer's advice and the detailed methodology was added to the revised version. (Line306-314)

18. Line 297-299 - Understandably, a powerful gene expression control system with many modifications to custom the gene control required. The authors have not informed the reader which approach of this system they used, neither do they cite work where the approach could have been used. Similarly, the complementation system, did the authors use it exactly as reported in the cited paper or did they adapt it to perform their work. Those details are lacking for others to be able to replicate these results should there be need to do so.

Reply: We appreciate for reviewer's advice and we have provided more detailed methodology in the revised version. (Line349-367)

19. Line 306-308. Perhaps this line should come last in this paragraph.

Reply: We appreciate for reviewer's advice and we have changed it. (Line379-380)

20. Line 315-316 -what groups are the authors referring to?

Reply: We thank the reviewer for carefully reading and pointing out the deficiencies of our manuscript. The groups here were referring the gene for qPCR (*argC*, *pyrP*, *nadA*, *gbuB*, *lmo2097*, *motB*, *rplB*, *flaA*). To make it clearer, we had rephrased this sentence. (Line384-386)

21. Line 458- labels 1, 2, 3 and 4 are missing on the figure.

Reply: Thanks for the reviewer's advice and we have added the labels on the figure. (Line562)

22. Line 464-465 -which X-axis are the authors referring to in Figure 1?

Reply: We thank the reviewer for carefully reading and pointing out the deficiencies of our manuscript. The height of column represents the survival number of the insertion mutants in BHI. We have corrected it. (Line571)

23. Fig 3 - What analytical software was used to generate the data plotted in this figure?

Reply: Thanks for the reviewer's question. The figure is drawn by 3D bubble chart in Excel. The x-axis represents the location of the gene in the chromosome, and the

y-axis represents the fold-change value (37°C versus 16°C). The bubble size represents the BH (statistical analysis) values.

24. Line 473 - these are not circles. they are spheres.

Reply: We appreciate for reviewer pointing out this error and we have corrected it. (Line580)

25. Line 474 - But why is the scale only up to -6 and +8, if change fold was e.g., 13 for lmo1366?

Reply: Thanks for the reviewer's question. The y-axis represents the Log₂FoldChange value (37 °C versus 16 °C), the Fold Change value varied from 2⁻⁶ (-64) to 2⁸ (256). The max Fold Change value for Tn-seq results was 76 (*cspL*). To avoid misunderstanding, we have rewritten the description of Fig.3 in the revised version. (Line579)

26. They have done Tn-seq and RNA-seq, however they do not mention if these have been deposited with any repository.

Reply: We thank the reviewer for their constructive point. Following the reviewer's suggestion, Illumina sequencing reads of Tn-seq (Accession No. SRR18767871- SRR18767875) and RNA-seq (Accession No. SRR18769210- SRR18769218) have been submitted in the NCBI server. (Line388-391)

Reviewer #4:

Wu et al. have used functional genomic analyses via Tn-seq to identify novel genes in *L. monocytogenes* involved in growth at low temperatures. The study methods, results and conclusions are sound and the findings of novel genes involved in bacterial growth at low temperature are notable. The paper would be greatly strengthened, however, by addressing the following:

Reply: We thank the reviewer for his/her compliments on our manuscript. Below we will reply in a point-by-point fashion to the reviewer's comments and the corresponding changes we have made in the manuscript.

1. The authors should describe Tn-seq principle in more detail in introduction, as the overall principle is not clear.

Reply: We appreciate the reviewer for this suggestion. Following the reviewer's suggestion, we have described Tn-seq principle in more details in the revised version. (Line84-90)

2. In the results, it is still unclear why exactly lmo1336 and lmo2301 were selected for further study. Please clarify why these looked to be among the most promising markers based on the data presented (ln 145).

Reply: We are grateful to the reviewer for pointing this out. The Tn-seq screening identified 79 genes contributing to low-temperature growth and 61 genes play negative roles for the growth at 16 °C. As shown in Fig.3, the genes (*cspL*, *bipA*, *pfkA*, *lmaB*, *rhlA*, *rhlD*) associated with low temperatures had been reported previously. And we chose lmo1366 and lmo2301 as candidate genes because they had significant fold change and have not been reported to be associated with low-temperature growth yet. To make it clearer, these explanations were added to the revised version. (Line163-166)

- The authors focus on temperature regulation and thermal survival pathways, but it's unclear whether these same pathways are truly specific to temperature or if pH, nutrients, etc. also play a role. It would be helpful to also look at morphology in milk cultures, for example. Is any of this data available?

Reply: We are grateful to the reviewer for pointing this out. The experiments designed in milk at different temperatures here is to see if the mutant of gene identified in BHI have similar growth trends in the real food system. We didn't put much effort into milk in this article. We will take reviewer's advice to look at morphology in milk cultures in another study "Functional genomics identified novel genes involved in growth in milk in *Listeria monocytogenes*".

- The authors suggest that findings could contribute to development of novel antibacterial strategies but don't elaborate on this in sufficient detail. How, exactly, could these findings translate to Lm sterilization or prevention strategies and what should be the priorities for future research and implementation efforts?

Reply: We appreciate for reviewer's advice. The relevant information has been added in the conclusion section and was described in a more modest way.(Line 290-293)

- Italicize genes throughout

Reply: We thank the reviewer for carefully reading and pointing out the error. We have italicized genes throughout.

- Please review throughout for punctuation and spacing errors in addition to language throughout, with special attention to past and present tense for consistency

Reply: We appreciate for reviewer's advice. Following the reviewer's suggestion, we have revised the manuscript to improve the English language by American Journal Experts, the English editing service recommended by ASM.

7. Ln49-50 Combine sentences
Reply: We appreciate for reviewer's comment and we have rewritten this sentence in the revised manuscript. (Line53-56)
8. Ln54: delete word "invasive"
Reply: We have corrected it. (Line23).
9. Ln 55: change to "immunosuppressed"
Reply: We have corrected it. (Line 60)
10. Ln 62: "novel food vehicles"- please explain or replace
Reply: We thank the reviewer's comment and we have re-written this sentence. (Line 68-72)
11. Ln 73: Sentence doesn't necessarily follow. Instead say that knowledge is incomplete
Reply: We thank the reviewer's advice and we have re-written this sentence. (Line 82)
12. Ln 166: Change to "complements"
Reply: We have corrected it. (Line 189)
13. Fig 3: Would suggest making background circles an even lighter color to reduce noise
Reply: We appreciate for reviewer's advice and we changed background circles to an even lighter color to reduce noise. (Line 566)
14. Fig 4: Make it clear how image was generated.

Reply: We appreciate the reviewer for the valuable suggestion. We have added “Protein prediction” in the Materials and Methods in the revised version. (Line 343-346)

15. Fig S1: Were any statistics applied here? Any significant findings?

Reply: We are grateful to the reviewer for pointing this out. The aim of this experiment was to probe the heat tolerant of the Δ Imo1366 and Δ Imo2301. The results showed no significant differences between the wild-type strain and the mutants (Fig.S1), suggesting the inactivation of Imo1366 and Imo2301 has no effect on the growth at high temperatures. (Line 208-211)

Reference

1. McGann P, Ivanek R, Wiedmann M, Boor KJ. 2007. Temperature-Dependent Expression of *Listeria monocytogenes* Internalin and Internalin-Like Genes Suggests Functional Diversity of These Proteins among the Listeriae. *Applied and Environmental Microbiology* 73:2806-2814.
2. Alegbeleye O, Sant’Ana AS. 2022. Survival and growth behaviour of *Listeria monocytogenes* in ready-to-eat vegetable salads. *Food Control*.
3. Rachon G, Peñaloza W, Gibbs PA. 2016. Inactivation of *Salmonella*, *Listeria monocytogenes* and *Enterococcus faecium* NRRL B-2354 in a selection of low moisture foods. *International Journal of Food Microbiology* 231:16-25.
4. Xanthiakos K, Simos D, Angelidis AS, Nychas GJ-E, Koutsoumanis K. 2006. Dynamic modeling of *Listeria monocytogenes* growth in pasteurized milk. *Journal of Applied Microbiology* 100:1289-1298.
5. Sheen S, Hwang C-A, Juneja VK. 2011. Modeling the impact of chlorine on the behavior of *Listeria monocytogenes* on ready-to-eat meats. *Food Microbiology* 28:1095-1100.
6. Gougouli M, Angelidis AS, Koutsoumanis K. 2008. A Study on the Kinetic Behavior of *Listeria monocytogenes* in Ice Cream Stored Under Static and Dynamic Chilling and Freezing Conditions. *Journal of Dairy Science* 91:523-530.
7. de Passos ALS. 1999. Identification of Genes Necessary for Growth of *Listeria Monocytogenes* at Low Temperatures. University of Leicester (United Kingdom), Ann Arbor.
8. Chen S, Meng F, Sun X, Yao H, Jiao X. 2019. Epidemiology of Human Listeriosis in China During 2008–2017. *Foodborne Pathogens and Disease* 17.

May 28, 2022

Prof. Xinglin Zhang
Zhejiang University
Yuhangtang Rd.866
hangzhou
China

Re: Spectrum00710-22R1 (Functional genomics identified novel genes involved in growth at low temperatures in *Listeria monocytogenes*)

Dear Prof. Xinglin Zhang:

as you will read from the comments below, while one reviewer was generally happy with your modifications, the second one is suggesting some more improvements in your manuscript. I must underline that if in the new revised version these improvements will not be achieved, so to satisfy the requests of the evaluators, I will not be in a position to accept for publication your manuscript.

Link Not Available

Sincerely,

Luca Cocolin

Journals Department
Reviewer comments:

Reviewer #3 (Comments for the Author):

Thank you for providing all the explanations to my queries.

Query 1. I beg to differ. Although the author provides supporting papers that have used 16oC for analysis. In my opinion research should as much as possible create conditions near to those that occur at the source of the problem. In this instance 4oC is achievable, but 16oC simply speeds acquisition of publication data and not understanding to solve the problem which occurs at 4oC.

Query 2. Here too, I beg to differ, because the Lm that causes problems grows at that temperature in association with the stress that the author is referring too. Basically, here it is a question of "seeing less rapidly" than "seeing more slowly", both are doable.

Reviewer #4 (Comments for the Author):

1. The authors have worked to address all comments and have even run their paper through the AJE editing service to improve the text. Unfortunately, it appears the AJE editing service was used before the edits and additions to the text, so the edited version should also be checked for English language and grammar.
2. In regards to the content of the revised submission, I agree with the rationale behind selecting 16C as opposed to 4C to select candidate markers, though the discussion behind this is still a bit unfocused within the manuscript itself. I would recommend the authors clearly state why 16C was selected as the study temperature in 1-2 lines- although references to studies of *L. monocytogenes* specifically at 16C have been added, it would be helpful to focus more generally on studies showing how less extreme temperature changes can be valuable to select such markers using Tn-seq (as per authors' response to the 3rd reviewer comment).
3. It remains unclear that the identified markers are definitively associated with temperature, as opposed to pH, light, nutrients, other factors. I appreciate the authors' response regarding additional studies currently underway to answer such open questions, and this limitation should also be added to the discussion of the manuscript.
4. Additionally, the conclusions in regards to public health implications of this and related work remain research-focused. A broader context of this work with the end goal of food pathogen control through scalable approaches is warranted.

Staff Comments:

Preparing Revision Guidelines

Please return the manuscript within 60 days; if you cannot complete the modification within this time period, please contact me. If you do not wish to modify the manuscript and prefer to submit it to another journal, please notify me of your decision immediately so that the manuscript may be formally withdrawn from consideration by Microbiology Spectrum.

Response to reviewers' comments

Dear Luca Cocolin,

On behalf of my co-authors, we thank you very much for allowing us to revise our manuscript, and we appreciate the editor and reviewers very much for their constructive comments and suggestions on our manuscript entitled "**Functional genomics identified novel genes involved in growth at low temperatures in *Listeria monocytogenes***".

We have studied the comments carefully and have made revisions which were underlined in the revised manuscript. We have performed additional experiments and tried our best to revise this manuscript according to the comments. Attached please find the revised version, which we would like to submit for your kind consideration. And responses to reviewers' comments are appended below.

We would like to express our great appreciation to you and the reviewers for their comments on our paper. Looking forward to hearing from you.

Yours sincerely,

Dr. Xinglin Zhang
Department of Food Science and Nutrition
Zhejiang University
Hangzhou 310058 P.R.China
Tel: +86-571-86984316
E-Mail: zhangxinglin@lyu.edu.cn

List of Responses.

Reviewer #3

Thank you for providing all the explanations to my queries.

Query 1. I beg to differ. Although the author provides supporting papers that have used 16°C for analysis. In my opinion research should as much as possible create conditions near to those that occur at the source of the problem. In this instance 4°C is achievable, but 16°C simply speeds acquisition of publication data and not understanding to solve the problem which occurs at 4°C.

Reply: We thank the reviewer for this constructive point. According to the reviewer's comments, we described the Introduction section more critically and appropriately and

integrated the reviewer's concerns into the Results and Discussion session (Line 68-70, 96-103, 203-204,230-232)

We would like to sincerely and humbly clarify that, at the beginning of this project, we also considered the 4°C condition and performed preliminary experiments. The reason for choosing 16°C is mainly because of the technical limitations of Tn-seq itself.

Unlike RNA-Seq or other genomic studies technologies, Tn-seq screening is based on a library containing large populations of different mutants (63666 unique insertion mutants in this study). These mutant strains do not exist in even proportions. When the screening conditions are too strict, their proportions might be deregulated randomly rather than reflecting the response to the screening conditions truly, which will lead to false-positive results in subsequent sequencing and statistical analysis. Moreover, extreme stress conditions would lead to identifying too many genes and generate a lot of 'noise', which will affect subsequent sequencing and bioinformatic analysis, eventually resulting in false positives or overwhelming the real key genes by the non-critical genes.

This is similar to the fact that when using mutant libraries to screen genes contributing to gastric acid resistance or antibiotic resistance, the actual pH value of gastric acid (1.5-3.0) and clinical concentration of antibiotics were often not used. Instead, milder conditions were often used for screening. For example, to identify genes involved in acid stress resistance of *Salmonella derby*, PH 4.0 was used for Tn-seq (1). And the MIC of daptomycin concentrations for *Staphylococcus aureus* was 2.5 µg/mL, lower concentration of daptomycin (0, 0.12, 0.25, 0.5, and 1µg/mL) were used for Tn-seq screening (2).

On the other hand, 16 °C as the research condition has practical significance. Many listeriosis outbreaks occurred due to poor food process environment and abusive storage temperature (3). The previous study reported that a listeria outbreak in the United Kingdom was associated with a poor food process environment (like contaminated chopping boards, sink plug holes, and cleaning sponges) (4). Furthermore, the environment temperature in food processing was usually well above 4°C. While production facilities have closely monitored temperature management systems, refrigerated display cases at the retail level and home refrigerators are often found at abusive temperatures (5). Although 4°C is the most common temperature for refrigeration, a national survey in the U.S. showed that the home refrigeration temperatures for ready-to-eat foods could range from below 0°C to15.6°C (6).

In the end, the reviewer's concerns provide us with a valuable point for our experimental setup in the follow-up studies. For instance, we could try to test more detailed and wide-range temperature conditions, including sub-zero repeated freeze-thaw survival test, and growth at 4°C, 16°C, 25°C, 37°C, 42°C. Although these experiments might be challenging for Tn-seq screening, we believe it worth a try and may provide important biological and biotechnological implications.

Query 2. Here too, I beg to differ, because the Lm that causes problems grows at that temperature in association with the stress that the author is referring too. Basically, here it is a question of "seeing less rapidly" than "seeing more slowly", both are doable.

Reply: We thank the reviewer' comment. We would like to sincerely and humbly clarify that, at the beginning of this project, we also considered the 4°C condition and did preliminary experiments. The reason for choosing 16°C is mainly because of the technical limitations of Tn-seq itself as we described above. In addition, listeria outbreak could be associated with a poor food process environment with temperatures above 4°C (4-6). The temperature 16°C was a trade-off between the realistic condition and the technical limitation of Tn-seq screening.

Although the targeted mutant strains showed consistent results at 4°C and 16°C, it did not represent that all the genes screened by Tn-seq at 16°C would necessarily get consistent results at 4°C.

Reviewer #4:

1. The authors have worked to address all comments and have even run their paper through the AJE editing service to improve the text. Unfortunately, it appears the AJE editing service was used before the edits and additions to the text, so the edited version should also be checked for English language and grammar.

Reply: We thank the reviewer for their constructive point. In this revised version, the English language and grammar were carefully corrected manually by “Grammarly” (a powerful paid writing assistant software for academic writing, www.grammarly.com). Meanwhile, we got help from native English speakers with the same research background. All the changes were underlined in the “Marked Up Manuscript”.

2. In regards to the content of the revised submission, I agree with the rationale behind selecting 16°C as opposed to 4°C to select candidate markers, though the discussion behind this is still a bit unfocused within the manuscript itself. I would recommend the authors clearly state why 16°C was selected as the study temperature in 1-2 lines- although references to studies of *L. monocytogenes* specifically at 16°C have been added, it would be helpful to focus more generally on studies showing how less extreme temperature changes can be valuable to select such markers using Tn-seq (as per authors' response to the 3rd reviewer comment).

Reply: We appreciate for reviewer's advice. Following the reviewer's suggestion, we added it in the revised manuscript and clarified to the readers that the temperature 16 °C was a trade-off between the realistic condition and the technical limitation of Tn-seq screening as we reply to the 3rd reviewer above. (Line96-103)

3. It remains unclear that the identified markers are definitively associated with temperature, as opposed to pH, light, nutrients, other factors. I appreciate the authors' response regarding additional studies currently underway to answer such open questions, and this limitation should also be added to the discussion of the manuscript.

Reply: We thank the reviewer for this constructive point. To determine whether *lmo1366* and *lmo2301* are associated with other factors, we performed the growth curve of mutants at different pH and osmotic pressure conditions. The results showed no significant difference between mutants and wild type, indicating that *lmo1366* and *lmo2301* are specifically associated with temperature. These results were added to the revised version (Fig.S4, Fig.5E). (Line225-229).

Fig. S4 Growth curves of the WT (●), Δ lmo1366 (▲), Δ lmo1366+comp (■), Δ lmo2301(▼), Δ lmo2301+comp (◆) in BHI at 37°C with PH (7.0, 6.0, 5.0, 4.5) and with NaCl (5%, 10%).

4. Additionally, the conclusions in regards to public health implications of this and related work remain research-focused. A broader context of this work with the end goal of food pathogen control through scalable approaches is warranted.

Reply: We appreciate for reviewer’s advice. The Conclusion was revised and sentences was added: “Collectively, these results reveal novel mechanisms of low-temperature growth in *L. monocytogenes* and may provide unique antimicrobial targets for the development of novel strategies to control food-borne pathogens in low-temperature storage or processing environments. Targeting rRNA methyltransferases can inhibit translation and eliminate the ability to methylate its substrates hence inhibiting the growth of *Lm* at low temperatures. In the future, applying theoretical results to food control through scalable approaches is the focus of our research, and more related research is warranted.” (Line312-318)

Reference

1. Gu D, Xue H, Yuan X, Yu J, Xu X, Huang Y, Li M, Zhai X, Pan Z, Zhang Y, Jiao X. 2021. Genome-Wide Identification of Genes Involved in Acid Stress Resistance of *Salmonella Derby*. *Genes* 12:476.
2. Coe KA, Lee W, Stone MC, Komazin-Meredith G, Meredith TC, Grad YH, Walker S. 2019. Multi-strain Tn-Seq reveals common daptomycin resistance determinants in *Staphylococcus aureus*. *PLOS Pathogens* 15:e1007862.
3. CDC. 2022. Listeria Outbreaks. <https://www.cdc.gov/listeria/outbreaks/index.html>. Accessed June 1st.
4. Dawson SJ, Evans MR, Willby D, Bardwell J, Chamberlain N, Lewis DA. 2006. Listeria outbreak associated with sandwich consumption from a hospital retail shop, United Kingdom. *Eurosurveillance* 11:632.
5. Ziegler M, Rüegg S, Stephan R, Guldemann C. 2018. Growth potential of *Listeria monocytogenes* in six different RTE fruit products: impact of food matrix, storage temperature and shelf life. *Italian journal of food safety* 7:7581-7581.
6. Pouillot R, Lubran MB, Cates SC, Dennis S. 2010. Estimating parametric distributions of storage time and temperature of ready-to-eat foods for U.S. households. *J Food Prot* 73:312-21.

June 4, 2022

Prof. Xinglin Zhang
Zhejiang University
Yuhangtang Rd.866
hangzhou
China

Re: Spectrum00710-22R2 (Functional genomics identified novel genes involved in growth at low temperatures in *Listeria monocytogenes*)

Dear Prof. Xinglin Zhang:

Your manuscript has been accepted, and I am forwarding it to the ASM Journals Department for publication. You will be notified when your proofs are ready to be viewed.

Sincerely,

Luca Cocolin
Editor, Microbiology Spectrum

Journals Department
Supplemental Dataset: Accept
Supplemental Material: Accept